# Melatonin: Facts, Extrapolations and Clinical Trials

**DOI:** 10.3390/biom13060943

**Published:** 2023-06-05

**Authors:** J. A. Boutin, D. J. Kennaway, R. Jockers

**Affiliations:** 1Laboratory of Neuronal and Neuroendocrine Differentiation and Communication, University of Normandy, INSERM U1239, 76000 Rouen, France; 2Robinson Research Institute and Adelaide School of Medicine, University of Adelaide, Adelaide Health and Medical Science Building, North Terrace, Adelaide, SA 5006, Australia; david.kennaway@adelaide.edu.au; 3Institut Cochin, Université Paris Cité, INSERM, CNRS, 75014 Paris, France; ralf.jockers@inserm.fr

**Keywords:** melatonin, clinical trials, controversies, preclinical data, mitochondria, bacteria, scavenging hypothesis

## Abstract

Melatonin is a fascinating molecule that has captured the imagination of many scientists since its discovery in 1958. In recent times, the focus has changed from investigating its natural role as a transducer of biological time for physiological systems to hypothesized roles in virtually all clinical conditions. This goes along with the appearance of extensive literature claiming the (generally) positive benefits of high doses of melatonin in animal models and various clinical situations that would not be receptor-mediated. Based on the assumption that melatonin is safe, high doses have been administered to patients, including the elderly and children, in clinical trials. In this review, we critically review the corresponding literature, including the hypotheses that melatonin acts as a scavenger molecule, in particular in mitochondria, by trying not only to contextualize these interests but also by attempting to separate the wheat from the chaff (or the wishful thinking from the facts). We conclude that most claims remain hypotheses and that the experimental evidence used to promote them is limited and sometimes flawed. Our review will hopefully encourage clinical researchers to reflect on what melatonin can and cannot do and help move the field forward on a solid basis.

## 1. Foreword

A review is a survey of the existing literature on a given subject and the analysis of it by the author(s). Even if the interpretation of the author(s) about the facts is important, it should be identified as such. The present review presents a critical assessment of the existing literature on melatonin effects, including the conclusions drawn by their authors. Such a critical assessment is, by nature, subjective and has to be further discussed by the community in order to reach a consensus on where the field stands in terms of its hypotheses and facts. The new consensual hypothesis and perspective should then guide future research, which will go through the usual cycle of data acquisition that will then be submitted for peer review.

## 2. Background, Facts

Melatonin is a natural compound synthesized in the pineal gland during the nighttime, discovered by Lerner et al. [1].

Melatonin is synthesized in the pineal gland under the control of the hypothalamic suprachiasmatic nucleus, such that it is high during the night and low during the day. In humans, circulating melatonin is lowest during the day, at <2 pg/mL (8 pM), and is at 30–70 pg/mL (130–300 pM) during the night [2]. Of course, absolute concentrations can be misleading, as they depend on the age, conditions and health status of the patients; however, these levels are generally accepted by the community and were also validated in post-mortem measures of melatonin in human pineal glands [3,4]. The biosynthesis of melatonin is well established [5,6]. This cycle commands the wake/sleep cycle of almost all animals [7]. As the day length varies during the year, melatonin also indirectly commands the circannual rhythm and thus the reproduction season [8].

The main targets of endogenous melatonin in mammals are its two G-protein coupled receptors, MT_1_ and MT_2,_ which have high affinities for the hormone 1 nM and lower. The receptors are activated by circulating levels of melatonin at night and are believed to translate most of its physiological actions [9].

## 3. Background, History

The history of melatonin use can be divided into two periods:

The first period is characterized by the establishment of a wide consensus about the nature of melatonin’s role, its synthesis and its regulation with outstanding discoveries on those subjects, and was followed by progress on the molecular mechanisms by which melatonin exerts its effect on biological rhythms [9] and sleep initiation [10,11]. In the meantime, the pharmaceutical industry described analogs of melatonin that serve as drugs in disordered sleep conditions [11,12], such as delayed sleep-wake phase disorder and jetlag, but also in cases where disrupted melatonin rhythmicity was associated with neurological conditions such as depression and anxiety [13], where they have been promising results based on standard clinical trials [14,15]. The present essay does not address this aspect of melatonin or its analogs (e.g., agomelatine, tasimelteon or ramelteon).

The second period of melatonin research is still ongoing and addresses the possible role of melatonin as an antioxidant and scavenger molecule; however, this is characterized by the absence of a general consensus in the field [16]. Hundreds of publications have proposed that melatonin can act as an agent to treat almost all the main disease conditions, such as cancers, obesity, Alzheimer’s, Parkinson’s and various virological problems, such as AIDS, Ebola and COVID-19 (see Table 1). We have previously reviewed several of these conditions, raising doubts about some claims [17,18], questioning whether a natural molecule could be used to treat so many diseases without actually addressing the mechanism(s) of actions or suggesting mechanisms of action that were independent of the melatonin receptors. Table 1 is mostly restricted to the more recent or significant claims. In short, this table essentially regroups the reviews in which the activity of melatonin on a given pathology is indicated/extrapolated. A common trait of most of these studies is the use of supra-physiological concentrations of melatonin (sometimes by six orders of magnitude higher than physiological concentrations). As pointed out earlier, “In principle, the established role of melatonin in rhythmic function is not necessarily incompatible with the use of high doses for ‘protective’ effects.” [19]; however, researchers need to carefully distinguish between the role of endogenous melatonin and the effects observed at high doses of exogenous melatonin. In a clinical setting, the question of specificity and side effects have to be carefully addressed at these supra-physiological concentration levels (see below).

## 4. Melatonin Goes to Clinic Why? For What?

In order to be entered into a clinical trial, a compound should have shown activity on the defined pathological model(s), exhibit low or no toxicity and be superior to the existing drugs (if there are any). Dietary “supplements”, as melatonin is defined in the USA, do not have the same legal constraints as pharmaceuticals, and their marketing information must not infer their use to treat diseases.

In general, such clinical trials are directed at a specific pathological condition and are based on a body of relevant preclinical studies. In the case of melatonin, it seems that almost all the pathological conditions one can think of have been reported in the literature to be ameliorated by melatonin! Thus, the conditions for which melatonin has been tested in clinical studies are extensive and are listed in Table 2 and Table 3. It is difficult to find a category that is not listed, and the immense spectrum of conditions reported to be treatable by melatonin is the obvious weakness of those claims.

In brief, whenever a clinical trial is launched, if declared on the official site (www.ClinicalTrials.gov, accessed on 23 January 2023, see below), the declaration comprises a series of keywords/categories, according to which the compound tested would have an effect—presumably—on the disease discussed (See Table 2 and Table 3 for an exhaustive list of such keywords linked to actual clinical studies, past, present and future).

One should recall (i) all the studies on a compound are not necessarily declared on this site, and (ii), more surprisingly, the data obtained are not necessarily published or made public. They belong to the initiator of the study and remain its property, translating into an absence of published results in most cases.

One should also point out that some of these studies are purely observational, aimed at measuring the levels of melatonin in patients with particular pathological conditions.

## 5. Clinical Studies

Melatonin is sold over the counter in several Western countries, such as Western Europa and the USA, in doses ranging from 1 to 10 mg. In clinical trials, doses of up to 100 mg have been reported (see Table 4). Several studies on humans showed that 100 mg of melatonin would result in a plasma Cmax of 1,252,500 pg/mL [i.v., bolus [39]] or 101 163 pg/mL [oral [40]]. Incidentally, 100 mg can be considered a huge dosage—the maximal level recorded in the complete review of Harpsøe et al. [41]. This raises the question of the safety of such doses. Because melatonin is considered a natural agent and not a drug, it is accessible without prescription in the USA, Canada and some other Western countries. There have been no large, long-term, high-quality randomized clinical trials specifically addressing melatonin safety in adults or children, probably because no Phase I information is required prior to commencing a clinical trial. It is unlikely that manufacturers and suppliers of melatonin as a dietary supplement would ever sponsor such an expensive trial. Nevertheless, melatonin is said to have a benign safety profile [42], yet this flies in the face of the multitude of physiological systems that melatonin has been associated with that are unrelated to any role in sleep. In a recent review, interactions with the cardiovascular, reproductive, endocrine and metabolic systems in humans were discussed, along with prescription drug interactions [43]. These interactions were discovered in controlled experiments in healthy subjects; we do not know what the effects of melatonin might be for people with cardiovascular diseases, diabetes, cancer, etc. A drug can only be termed safe in relation to what has been investigated. Nevertheless, even in 2023, researchers are calling for studies on the long-term safety of melatonin [44,45,46]. In the meantime, we are left with a somewhat patchy data collection via poison centers receiving information on adverse events (www.poison.org/ accessed on 4 February 2023). Indeed, the review by Vines et al. [47] indicated that no formal safety trials had been performed. All the available information in the public domain comes from poison and other centers, as briefly summarized here: “Health Information” given by the NIH on the side effects of melatonin mentions headaches, dizziness, nausea and sleepiness, with the possible long-term side effects remaining unclear [48]. Furthermore, a recent report by the French Agency for Food, Environmental and Occupational Health and Safety (ANSES) reached a similar conclusion based on 90 responses collected in a nutri-vigilance program between 2009 and 2017 [49], which was complemented by similar data available from other European countries and Canada [50]. Of note, the alert put forward a possible link between melatonin and infant sudden death syndrome [51]. While this study could not directly attribute these infant deaths to the use of melatonin, they concluded that “Melatonin is not known to be acutely toxic; however, it causes a multitude of systemic effects by way of mechanisms of action that are not entirely understood, especially in developing infants.” More guidance is critical for parents and pediatricians who use melatonin as a routine sleep aid for young children. This caution is also shared by another group in their recent communication [52] (see also Section 6).

Interrogating the www.ClinicalTrials.gov site (accessed on 23 January 2023), we found an impressive number (742) of clinical studies on melatonin on the conditions listed in Table 2 and Table 3. The condition list can be divided into two parts: the “generic” ones; these 399 conditions regroup several trials (Table 2), for example, “Mental disorders” containing 279 studies, such as “psychosis & sleep”, “Rapid Eye Movement, Sleep Behavior Disorder & Parkinson Disease” etc., and the “unique” ones (296), corresponding to a single study, such as “Digestive system diseases or Necrotizing Enterocolitis” or “Bacteremia or Human Endotoxaemia”, etc. (Table 3). By tapping the conditions into the search query on the site, one can have access to the trials and their details.

This body of trials comprises all the declared trials in the past, present and future. As a basis for comparison, we did the same query for “resveratrol”, a natural product also reported having many beneficial actions in traditional medicine, and found 147 studies, or “vitamin C”, and found 453 trials (400 of which are completed). Melatonin has attracted a lot of attention, as the diversity of conditions (and thus of pathological conditions) covers most, if not all, human diseases.

From the trial list, we have removed 118 conditions where the status is “unknown”. Another 40 studies were reportedly withdrawn, without obvious reasons, but one might speculate that this was due to difficulties in recruiting patients or to the lack of results, particularly on delirium prophylaxis, Parkinson’s disease, coronary artery calcification, intensive care elderly population or “melatonin inhibition of NLRP3 inflammasome in COVID19 patients”, to name only but a few.

Looking at the 56 trials currently recruiting, one can separate them into two main categories: observational and interventional. The first section, “observational”, comprises all the studies in which the patient’s melatonin levels are measured as a function of (i) their pathological conditions and (ii) their daily circadian rhythms. Those studies are basically clinical biochemistry research trials and are extremely useful for our understanding of the dependence of circadian rhythms on pathological conditions. Among the pathological conditions that have been studied are Cushing’s syndrome, autism, hypo-hidrotic ectodermal dysplasia, bipolar disorder, frail elderly, etc. The data arising from those studies may be expected to have a major impact on our understanding of how melatonin synthesis and release are influenced by pathological conditions. In our view, those studies are quite important, hopefully leading to scientific paper(s) explaining the results, as those in the Harpsoe et al. review [41], including the negative results in which the melatonin rhythm remains completely “normal”.

The intervention group comprises all the studies in which melatonin is tested as a therapeutic agent. Table 4 lists 35 recruiting studies, which are both interventional and observational. It is clear that the conditions studied are numerous and diverse, including cognitive dysfunction, bone diseases, attention deficit and disruptive behavior disorders, bone fractures, chorea, coronavirus infections, diabetes complications, post-operative pain, arthritis and traumatic brain injuries.

Because cancer is one of the most documented and searched domains in pathology, we have selected the trials in which cancer patients were treated with melatonin. Sixty-five studies tested its effectiveness in various cancer-related parameters, such as a loss of appetite, sleep quality, etc. Those studies also comprised trials in which the effects of melatonin as an anti-cancer adjuvant or drug, with direct effects on the disease itself, were reported. Half of the trials are reported as completed (32 studies), yet only the following four had published their results. We have briefly summarized their results and their conclusions:

Trial NCT00513357 (USA): Conclusion: “In cachectic patients with advanced cancer, oral melatonin 20 mg at night did not improve appetite, weight, or quality of life compared with placebo” [53].

Trial NCT00668707 (Canada): Conclusion: “Melatonin may benefit cancer patients who are also receiving chemotherapy, radiotherapy, supportive therapy or palliative therapy by improving survival and ameliorating the side effects of chemotherapy” [54].

Trial NCT00925899 (Denmark): Conclusion: “In the current study, oral melatonin at a dose of 20 mg was not found to improve fatigue or other symptoms in patients with advanced cancer” [55].

Trial NCT04137627 (Indonesia): Conclusion: “In patients with squamous cell carcinoma of the oral cavity, the addition of 20-mg melatonin to neoadjuvant chemotherapy reduced the expression of miR-210 and CD44 and decreased the percentage of tumor residue; however, no statistically significant result was observed”.

The survey of those trials currently recruiting is a good image of the other ongoing trials: from those using small to large dosages of melatonin, 50 to 100 patients, spanning a month to more than a year. We would argue that the spectrum of diseases and/or conditions cannot seriously be attributed to a single melatonin-related cause.

It is very often under-regarded that melatonin has poor bioavailability. Indeed, the vast majority (>80%) of melatonin is hydroxylated by cytochrome P450 [56], conjugated with sulphate or glucuronic acid and excreted as conjugates [57], which is a classical scheme in drug metabolism. Melatonin is also a substrate of the indoleamine 2,3-dioxygenase, leading to the opening of the indole cycle, leading to AFMK (N1-acetyl-N2-formyl-5-methoxykynurenamine), which is then rapidly metabolized further to AMK (N1-acetyl-5-methoxykynurenamine) [58].

The wide range of dosages for the patients undergoing these recruiting trials varies from 0.5 to 100 mg, with most of them in the low milligram range. Once again, one should recall that at low ‘physiological’ doses—those in the 1 to 3 mg range, the resulting C_max_ is low (~10 nM), even if it is two orders of magnitude higher than the nocturnal produced level. If one is given 100 mg, the circulating concentration of the compound is in the 0.5 µM range [40], as reviewed by Harpsoe et al. [41], although both concentrations vary widely between the various studies. Furthermore, the first pass principle of drug metabolism [59] clearly states that roughly half of the dose in the blood is eliminated by the liver minutes after ingestion and even shorter if injected intravenously [60]. This principle translates into the fact that melatonin remains in the µM range concentrations in the blood for a few minutes.

In the preclinical studies upon which trials are usually based, melatonin was generally used at concentrations higher than 1 µM—and even 1 mM. Thus, those concentrations are not expected to be naturally reached in living animals or in humans, and certainly not for prolonged periods. Thus, the transposition to clinical trials of those data might be considered more an act of faith than an experimental reality.

The main issue to be considered here is that nearly 4000 patients have entered these trials and have been treated with 0.5 mg to 100 mg melatonin daily. Furthermore, some of those studies are designed to run for very long periods (for several years, in some cases). In one clinical trial on melatonin and autoimmune diseases (ClinicalTrials.gov Identifier: NCT03540485, accessed on 23 January 2023), it is planned that 50 patients recruited because of their neurological conditions will receive a daily administration of 100 mg of melatonin, or a placebo, orally between 10 pm and 11 pm for 24 months. Finally, several studies are using newborn children (from a few hours to a few days old) with quite desperate health conditions—a feature that is commented on below.

In a survey (meta-analysis) of melatonin used for cardio-protection, the authors stated that the data was difficult to generalize due to the numerous possible biases, the low number of patients and their high heterogeneity [61]. Note that the combined number of patients was 396, half of which were tested with a placebo across six studies. Three of these studies were labeled as ‘no or poor effect’ (of melatonin as cardio-protectors). The doses were from 12 to 50 mg orally plus 1 or 2 mg intracoronary. Finally, the only study where the myocardial IR injury site was assessed using resonance imaging was classified as ‘no effect’ [62,63]. Incidentally, one of those articles stated, “…treatment with melatonin was associated with a larger infarct size in the group of patients” [63].

In conclusion, many clinical studies appear to have been based on scientifically poor experimental (preclinical) studies or overenthusiastic interpretations of the preclinical experimental data, thus explaining the overall poor outcome of these studies. The use of small populations of patients—elderly people as well as infants and toddlers in desperate health conditions—may have ethical considerations.

## 6. Melatonin in Kids

The first reports of the administration of melatonin to children were conducted by Jan and colleagues more than 28 years ago [64,65,66] in attempts to modify their disordered sleep resulting from neurodevelopmental disorders or blindness. Since that time, there have been many studies published on the pediatric use of melatonin, generally in children with comorbidities, such as autism spectrum disorder or attention deficit/hyperactivity disorder [44], as opposed to typically developing children with sleep problems. The majority of studies conducted in both adults and children have been small and short-term, although there are reports of treatments continuing for many years outside of the trials. The melatonin used in trials has usually been an immediate release preparation from non-pharmaceutical companies (dietary supplement manufacturers or chemical suppliers), with the first pharmaceutical preparation, Circadin, being approved in Europe in 2007. Circadin, a prolonged-release formulation, is authorized for the short-term treatment of primary insomnia in adults aged 55 years and older. It was not until 2018 that a formulation, Slenyto, was approved in Europe for the treatment of insomnia in children from 2 years of age with autism spectrum disorder and Smith–Magenis syndrome. With respect to the efficacy of melatonin in improving sleep in children presenting with sleep disorders, there is evidence that melatonin can have a role in those with neurodevelopmental disorders [42,67,68]; however, the effects are modest [43].

Since melatonin in the USA is not considered a drug, there is no requirement to report adverse events. The latest such report, covering 2012–21, documented more than 250,000 pediatric ingestions, 45,000 symptomatic effects, 3211 serious outcomes and 2 deaths [69]. Further, unintentional melatonin ingestion and its related hospitalizations and serious outcomes are increasing in the USA [69]. A separate recent study reported seven undetermined deaths of infants and toddlers with high exogenous melatonin as a result of deliberate or incidental ingestion of melatonin [51]. Based on these observations, careful evaluation and caution might be wise before administering melatonin to young children.

It is true that one cannot be sure that melatonin played a significant role in the deaths of infants. It is, however, an indication that melatonin is being recklessly administered to children in the absence of any formal safety information. Even the pharmaceutical companies that sell melatonin say not to take melatonin during pregnancy or when breastfeeding.

The uncertainty about the use of (high dosages of) melatonin by mothers during pregnancy and lactation remains. Indeed, in their thorough review of the subject, Vine et al. [47] concluded, “clinical studies to date suggest that melatonin use during pregnancy and breastfeeding is probably safe in humans and emphasizes the need for clinical studies…, including exogenous melatonin, during pregnancy and lactation”. The key word, for us, is certainly “probably safe”. To the best of our knowledge, no actual trial results whose primary outcomes were the safety or efficacy of melatonin for insomnia or other sleep disorders during pregnancy and lactation have been issued. Furthermore, no trial comprising “lactation” has been declared on the trial site.

Overall, our feeling is that, while the study could not directly attribute the deaths to the use of melatonin, they concluded that “Melatonin is not known to be acutely toxic; however, it causes a multitude of systemic effects by way of mechanisms of action that are not entirely understood, especially in developing infants. More guidance is critical for parents and pediatricians who use melatonin as a routine sleep aid for young children” [51].

## 7. Some Proposed Mechanisms behind the Claimed Beneficial Effects of Melatonin

### 7.1. Melatonin as an Antioxidant

#### 7.1.1. Melatonin as a Scavenger

In an initial influential paper, melatonin, as well as other indole-based molecules, was shown to be a good scavenger in an a-cellular system. Then, to render the experiment transposable to a “living” system, the tissue homogenate was added to the medium, and it was found that the trapping operation still occurred. Thus, melatonin was called a scavenger. As a direct consequence, more than 1727 (23 February 2023) articles have been published using the words “melatonin” and “scavenger”. The initial publications can be pinpointed to Poeggeler et al. [70] and. The influence of this statement can be seen through the evolution of the number of papers appearing in the literature over the years after this publication (Figure 1A). The possibility that melatonin could be an antioxidant molecule appeared early in 1958. Ever since this date, publications formulating hypotheses or reporting experiments started to appear in the literature. On 25 March 2023, a query of “melatonin” + “antioxidant” returned 24,271 results (see Figure 1B).

The statement that melatonin and many other purported antioxidants are scavengers is common in the introduction of hundreds of publications. We consider the term “scavengers” to be misleading because, while these compounds can react with oxidants, they are generally no more reactive than the thousands of other compounds in cells [71]. Thus, to be 50% effective, they would need to be at a concentration approximately equal to all the other compounds in the cell that could react with the oxidant. In cell culture experiments, scavenging can seem effective; however, it is likely to be because the ratio of the compound to its putative targets is likely thousands of times higher than would occur in vivo [72].

Scavengers do not exist in complex media, living systems, etc. A true antioxidant defense is provided by enzymes that remove superoxide and hydrogen peroxide with rate constants that are hundreds of thousands to more than a million times greater than that of melatonin. For hydroxyl radical, no compound is an effective scavenger, and the antioxidant defense involves the prevention of its formation rather than its removal (see discussions in [73,74,75]).

Many of the compounds purported to have antioxidant activity can react with Kelch-like erythroid cell-derived protein with CNC homology-associated protein 1 (Keap1), sparing Nrf2 from degradation and thereby activating transcription of the genes for antioxidant enzymes. Several compounds have been reported as such protectors for Keap1 degradation: TBE-31 [76,77], CDDO, a triterpenoid [78], or curcumin and caffeic acid [79,80], to name only but a few. However, many of those compounds also affect other signaling pathways that result in the protection of cells from injury [73].

#### 7.1.2. If Not a Scavenger, Then What?

The toxicity of ROS and oxidative stress is controlled by several enzymes, the activity of which detoxifies ROS, such as the three superoxide dismutases (see discussion in Keller et al. [81]. These enzymes are under the control of oxidative response elements that, once activated, induce their expression. This process is dependent on a nuclear receptor, such as nrf2, and melatonin has been suggested to be able to induce the expression of cellular defenses against oxidative stress (see also discussion in Amoroso et al. [82]). Many years later, this nuclear receptor has still not been found. Nrf2 might be a candidate [83,84], as might be Hem oxidase 1 [85], or both [86]; however, their role is yet to be experimentally demonstrated (see also Section 7.2).

### 7.2. Melatonin Nuclear Receptor

The proposed potential role of the nuclear receptors, promoted by many authors, remains an example of the enthusiastic support of a study that has not been reproduced (and thus confirmed) anywhere. In 1994, it was reported that melatonin binds to the receptor RZRα and activates it [87]. Another paper from the same group was published at the same time, reporting the binding affinities of iodo-melatonin in nuclear homogenates from cells in which RZR was cloned and expressed [88], which has been cited ever since 314 times. After several years of unsuccessful attempts to confirm the results in several laboratories, the 1994 paper was withdrawn in 1997 [89]. It has been cited and was continued to be cited 546 times, including 271 times since it was retracted. One possible explanation for that might be the seductiveness of the idea of a nuclear melatonin receptor, knowing that melatonin easily penetrates cells and nuclei, and the number of reported transcriptional effects of melatonin. The goal of the community is to discover the nuclear factor that is activated by melatonin leading to the neo-synthesis of antioxidant proteins, possibly explaining the melatonin antioxidant properties. The possibility that melatonin could be an antioxidant molecule appeared early in the 1990s [90]. Ever since this date, experiments and hypotheses publications started to appear in the literature; as a query, “melatonin” + “antioxidant” returned 24,271 results on 25 March 2023 (see Figure 1B).

### 7.3. Melatonin in Bacteria and Mitochondria

The idea that bacteria produce melatonin has been formulated in statements such as the following one: “Evolution’s best idea” is that melatonin is supposedly produced in bacteria. From there, because bacteria colonized cells several million years ago, eukaryotic cells became mitochondria; mitochondria are thus full of melatonin. Additionally, melatonin is there to protect mitochondria, cells and living organisms from oxidative stress [91].

#### 7.3.1. Melatonin in Bacteria

This statement can be challenged as follows:

Melatonin synthesis in microorganisms has been reviewed by Que et al. [92]. Indeed, numerous yeast, protozoa and algae are reported as capable of synthesizing the hormone, as optimistically reportedly reviewed by Hardeland et al. [93,94]. Algae, protozoa and fungi are eukaryotic organisms. On the contrary, bacteria are prokaryotic organisms, and they might have the capacity to synthesize melatonin, as reported in just six publications, listed in Table 5.

It is interesting to look closely at those articles cited to provide **proof** of bacterial melatonin synthesis.

In an article by Byeon and Back on the cloning of *E. coli* using the necessary enzymatic machinery for this bacterium to produce melatonin, the only mention of two bacterial melatonin productions is in the following sentence: “Melatonin is predicted to have evolved from the precursor bacteria of mitochondria and chloroplasts, such as *Rhodospirillum rubrum* and *Cyanobacteria*, respectively, via an endosymbiotic event with their ancestral eukaryotic host” [95]. This is not evidence of these bacteria producing melatonin.Manchester et al. reported a melatonin immunoreactivity in this *Rhodospirillum rubrum* [96]; however, this observation has never been reported independently, nor has melatonin been directly measured in the bacterium. Given the capital importance of *Rhodospirillum rubrum* and other purple non-sulfur bacteria, predicted to be at the origin of mitochondria according to the endosymbiotic theory of mitochondria [100], it is surprising that no more effort has been spent to clarify this point.Jiao et al. [97] reported a thorough study on eight strains of bacteria and found melatonin in only four: *Agrobacterium tumefaciens*, *Bacillus amyloliquefaciens*, *Bacillus thuringiensis* and barely in *Pseudomonas* sp.A review by Tan et al., while not providing any experimental data on bacteria melatonin production [98], stated that a series of *Lactobacillus* had been patented for their production of melatonin. Furthermore, they wrote: “If these speculations are valid, the beneficial effects in consumption of these products may, at least partially, be explained by the presence of melatonin and its isomers” [98]. This statement was later used as evidence that several *Lactobacillus* produce melatonin. For example, in a trial on abdominal pain in children, *Lactobacillus* and exogenous melatonin were used for the health of the patient [101], as if melatonin produced by *Lactobacillus* was not enough to obtain the desired effect. In another study, melatonin was shown to increase the amount of *Lactobacillus* previously decreased by sleep deprivation as if the production of melatonin by *Lactobacillus* was not enough to counterbalance the decrease in circulating melatonin [102]. Finally, in an earlier study on the possible relationship between several “antioxidant” molecules and their capacity to inhibit bacterial growth in the presence of mycotoxins, melatonin did not show any significant effect on the growth of *Lactobacillus* [103]; however, if this bacterium produces melatonin, those authors should have seen an effect of the addition of melatonin to the bacteria. In conclusion, the situation of *Lactobacillus* as a source of melatonin production remains, at the very least, unclear.In *Erythrobacter longus*, Tilden et al. [99] reported the presence of melatonin by using a radioimmunoassay, although there is reason to be cautious when using immunoassay melatonin by radioimmunoassay in complex matrices [2,104].

Very recently, Chen et al. reported the discovery of the gene encoding for a serotonin *N*-acetyltransferase gene, xoSNAT3, in the bacteria *Xanthomonas oryzae* [105]. The synthetic melatonin capacity of this bacterium was not reported in this paper, nor was the presence of melatonin. This very large family of enzymes has been reported from many organisms [106], including by us [107,108]; however, its expression does not grant the capacity to synthesize melatonin.

We conclude that whether bacteria widely produce melatonin remains to be proven.

#### 7.3.2. Melatonin in Mitochondria

Melatonin is claimed to be present/enriched in mitochondria [91,109].

However, robust and, in particular, quantitative data are not available. To our knowledge, massive amounts of melatonin have not been reported in mitochondria. Furthermore, if melatonin is a ROS scavenger, it would be expected to be transformed into hydroxy-melatonin or possibly into a kynurenamine, as shown by mass spectrometry analyses [110,111]. To the best of our knowledge, the generation of such metabolites was never directly correlated with the trapping of ROS by melatonin in mitochondria—or in other systems. However, 2-hydroxymelatonin, 4-hydroxymelatonin and 6-hydroxymelatonin have been reported in various other situations. Further examples are 2-hydroxymelatonin in the signaling pathways in Arabidopsis [112], the generation of 6- and 2-hydroxymelatonin upon the action of cytochrome P450 on melatonin [113] or the a-cellular system scavenging property of 4-hydroxymelatonin, which was described as superior to that of 2-hydroxymelatonin [114].

It is assumed that the mitochondria occupy a tenth to a twentieth of the total cellular volume [115]. Therefore, any concentration of melatonin in the mitochondria would translate as its tenth or twentieth volume in the whole cells. Assuming a high concentration of melatonin in mitochondria of 1 mM for its role in neutralizing ROS production, it would lead to cell homogenates in a melatonin concentration of ca. 50 to 100 µM (~12 to 23 µg/mL). A feature never measured in the literature is where, in blood, the melatonin concentration during the night is about 60 to 70 pg/mL [116], up to 300 pM. Similarly, in sheep brain tissue, melatonin concentrations of ~1 nM (i.e., 232 pg/mL) and 10 nM have been measured during the day and night, respectively [117]. This would translate into a concentration 10 times higher in mitochondria: 10 to 100 nM, which is far from any capacity to protect those organelles from ROS injuries. One can argue that melatonin concentrations are not in equilibrium between mitochondria and the rest of the cell because mitochondria have the capacity to either enrich melatonin or retain the melatonin synthesized in mitochondria. However, for the moment, there is no experimental evidence supporting these hypotheses. The well-characterized physical properties of melatonin show that exogenous melatonin equilibrates within seconds with the cytoplasm, confirming that melatonin crosses biological membranes rapidly [118].

The question of intra-mitochondrial melatonin synthesis has been recently reexamined by Suofo et al. [119]. The authors identified, using a Western blot, the two enzymes of the biosynthesis pathway, AANAT and ASMT, in preparations of purified mouse non-synaptosomal brain mitochondria. These bands were fully resistant to proteinase K digestion and digitonin treatment, indicating their localization in the mitochondrial matrix. In contrast to the pineal AANAT levels, these mitochondrial AANAT levels did not change along the circadian cycle. The authors demonstrated further that, in mouse neuroblastoma (N2a) cell knockout for AANAT, mitochondrial-generated superoxide production was increased. No alteration of the mitochondrial membrane potential was seen in these cells. In an attempt to address melatonin synthesis in mitochondria more directly, purified mitochondria were treated with deuterated (d4)-serotonin, the AANAT substrate. The generation of d4-n-acetylserotonin and d4-melatonin was observed by mass spectrometry, indicating that mitochondria can indeed synthesize melatonin if the serotonin precursor is available. These data represent, by far, the most convincing dataset in support of melatonin synthesis in mitochondria. However, several important questions still have to be solved. Furthermore, as whole-brain mitochondria were used, the question of whether melatonin synthesis can occur in the mitochondria of all brain cells or of a subset of cells is unknown. Whether this capacity of melatonin synthesis also exists in mitochondria from other tissues remains to be studied, as the quantity of mitochondria varies widely from cell type to cell type, and the repertoire of proteins imported into mitochondria largely depends on the cell type. Importantly, whether the generated quantities of melatonin are significant or anecdotal remains an open question, as no quantitative conclusion can be drawn from the data presented by Suofo et al. [119]. To show that melatonin is really synthesized by mitochondria in vivo, in other words, whether the precursor concentrations and enzyme levels are sufficient, remains an important goal for future studies.

#### 7.3.3. Conclusions

In summary, evidence on melatonin synthesis in bacteria remains weak and warrants independent replications. Other bacterial strains should be investigated to determine how generalizable they are. Studies should focus on *Rhodospirillum rubrum* and related species to provide evidence for the assumption that mitochondria are producing melatonin because the bacteria engulfed millions of years ago already produced this molecule. Collectively, the experimental evidence for melatonin synthesis in mitochondria and, in particular, its quantitative aspects and generalization to all mitochondria-containing cells is still weak. This includes the presence of its main precursors, such as serotonin, as well as its main metabolites that could be expected, assuming the highly oxidative environment of the mitochondrial matrix.

### 7.4. Melatonin as a Co-Substrate of NQO2

Although not really related to the previous points but providing an example of how ideas may emerge, a third melatonin receptor was reported [120] and systematized later on, particularly with a specific ligand, MCA-NAT [121,122]. The identity between this third melatonin binding site and the enzyme NQO2 has been reported [123,124]. In the following years, a mini-review was published hypothesizing that melatonin is, in fact, a co-substrate of NQO2 [125]. That would explain how melatonin regulates the antioxidant capacity of the enzyme. This hypothesis was then experimentally tested by another group and proven to be wrong [126]. Despite the disapproval of the initial hypothesis, it still continued to be mentioned in the literature by citing the mini-review with some citations, even transforming the hypothesis into a fact. Unfortunately, this persistence of the wrong hypothesis may have retarded the field by redirecting future research in the wrong direction.

## 8. Melatonin Has Many Effects and Targets

The suspected beneficial effect of melatonin in such a large spectrum of diseases is often justified by the involvement of melatonin in a large spectrum of biological processes, as reported in preclinical (cellular and animal) studies. To illustrate this diversity in terms of the melatonin effects, the following 30 randomly selected examples were extracted from a PubMed interrogation with the words “melatonin” and “inhibit”, which retrieved 400 items in total between 2018 and the end of 2022.

In also the following publications, the effect of melatonin is associated with a molecular target (IGF1, CYP, Akt, etc.) in a way that might suggest to the reader that melatonin is indeed binding to those targets. This is not the case; in most of the publications, the authors suggest that melatonin does “something” that leads to a pathway acting through one of these proteins without convincingly demonstrating a molecular interaction between melatonin and a protein. Melatonin protects from experimental models of newborn hypoxic-ischemic brain injury via the MT_1_ receptor [127]. Melatonin inhibits LH + insulin-like growth factor 1 (IGF1)-induced androstenedione and progesterone production as well as the expression of steroidogenic acute regulatory protein (StAR) mRNA (via real-time polymerase chain reaction) in Theca cells. Melatonin has no effect on cytochrome P450 11A1 (CYP11A1) and cytochrome P450 17A1 (CYP17A1) mRNA abundance [128]. Melatonin increases apoptosis, induced by cisplatin, by inhibiting the JNK/Parkin/mitophagy axis [129]. Melatonin elevates α-ketoglutarate and diverts adipose-derived exosomes to macrophages in mice [130]. Melatonin downregulates the expression of the dynamin-related protein 1 [131]. Melatonin restores mitochondrial normalcy after MPTP treatment in zebrafish [132]. Melatonin (3 mM), combined with 20 nM of rapamycin, suppresses the AKT/mTOR pathway activation, mitophagy and apoptosis via the regulation of mitochondrial function(s) in cultured cells [133]. Melatonin inhibits ERK phosphorylation [134]. Melatonin attenuates lung ischemia-reperfusion injury by inhibiting oxidative stress and inflammation [135]. Melatonin inhibits the HIF-1α-VEGF pathway in oxygen-induced retinopathy mice [136]. Melatonin activates Src and PKA in parallel and, thus, regulates CRE-dependent gene transcription [137]. Melatonin preserves insulin secretion and hepatic glycogen synthesis in rats [138]. Melatonin inhibits Cav3.2 T-type Ca^2+^ channels (about 40% at 10 µM) [139]. Melatonin activates the ERK1/2 signaling pathway [140]. Melatonin preserves the YAP expression during doxorubicin-induced cardiotoxicity [141]. Melatonin induces apoptosis in VCR-resistant oral cancer cells [142]. Melatonin induces apoptosis in 3T3-L1 preadipocytes as well [143]. However, melatonin inhibits apoptosis through the upregulation of sestrin2 in vascular smooth muscle cells [144]. Melatonin attenuates Atg5-dependent autophagy and activates the Akt/mTOR pathway [145]. Melatonin inhibits excessive mitophagy through the MT2/SIRT3/FoxO3a signaling pathway in cells [146]. Melatonin blocks the ROS-mediated HIF-1α/miR210/ISCU axis activation [147]. Melatonin inhibits the mitochondrial permeability transition pore opening [148]. Melatonin partially inhibits the NE/AKT/β-catenin/SLUG axis [149]. Melatonin inhibits TRPV4 activity (about 80 % at 1 mM) [150]. Melatonin promotes endocytosis and the subsequent degradation of HER2 [151]. Melatonin (0.2 mM) suppresses O-GlcNAcylation of cyclin-dependent-like kinase 5 [152]. Melatonin activates the ATF6 and PERK signaling pathways [153]. Melatonin activates the Nrf2/HO-1 signaling pathway [154]. Of note, the interaction between melatonin and NF-κB has been reported in numerous pathological cases, such as osteoarthritis [155], breast cancer [156] or, more generally, in inflammatory pathways [157]; however, the nature of the interaction has not been convincingly demonstrated.

This snapshot of articles is not meant to be exhaustive, as the search using keywords will always return only a partial picture. Nevertheless, we found the exercise quite informative on two counts:(i)The concentration of melatonin needed for most of the abovementioned effects is high, e.g., in the range of 1µM and up to several mMs. This leads to two remarks: (1) whether these effects are specific for melatonin or would they also be observed in structurally related molecules; (2) related to this first point is the question of the molecular target(s) or mechanism(s) behind it. Most studies lack the appropriate experimental conditions to—or at least start to—address these questions appropriately. For the first point, the most obvious structurally related candidate class of molecules is indoles (see below, Section 9); however, other classes of chemical compounds should also be considered, such as primary amines, etc. The second remark on molecular targets and mechanisms is a crucial step toward a better understanding of the observed effects [158]. The most obvious experiments, in this respect, are the establishment of concentration (dose)–response curves to determine an EC_50_ value. Low EC_50_ values generally hint at specific molecular targets, whereas a high EC_50_ value hints at targets with lower specificity. The absence of EC_50_ values (no saturation) may hint at a general property, such as membrane fluidity or intactness. Unfortunately, many of the articles describing the effects of melatonin use only a single (often high) melatonin concentration/dose.(ii)The reported effects of melatonin tend to be over-interpreted. The effects are not only system-dependent but, on their own, do not necessarily allow for conclusions on the precise mechanism or targets involved. These ‘descriptive’ data should therefore be interpreted with caution, not only in terms of the molecular mechanism and specificity but also in terms of translatability into another cell type, tissue and its relevance for pathologies. Unfortunately, the often-used perspective phrase “… these findings open new avenues in therapeutics” should be used with more precaution, considering the high melatonin concentrations used and the fact that almost all experimental protocols use melatonin in a preventive paradigm instead of a treatment paradigm.

Another misleading habit is the transformation of the working hypotheses and ideas formulated in an article into a fact in the following article and then propagated from review to review. This is nicely illustrated by the literature associating melatonin with the recent COVID-19 pandemic. A PubMed search using the terms “melatonin” and “COVID” retrieved 217 publications (25 February 2023) in just a three-year period. Incidentally, most of the papers claim that melatonin is active because it is an antioxidant and scavenger and has anti-inflammatory properties [159], and thus it is unsurprising to reach the conclusion that melatonin would be the ideal anti-COVID-19 therapeutic agent, as proclaimed in many reviews. However, a closer look at the more than 200 publications on COVID-19 and melatonin shows that most of them are reviews or comments and that experimental pieces of evidence are actually only reported in a small number of original articles. Interestingly, the anticipated anti-inflammatory effect of melatonin was not observed in K18-hACE2 mice infected with SARS-CoV-2 [160]. Whether this lack of effect is specific to this mouse model and that of a rapid manifestation of severe COVID-19 symptoms occurred remains to be determined in further studies. The totally unexpected effect of melatonin in this mouse model was the protection of the brain from SARS-CoV-2 infection, as compared to the lungs at high doses of melatonin [161]. Even more unexpected, the mechanistic studies suggest that this effect is mediated by the binding of melatonin to an allosteric binding site at the human angiotensin-converting enzyme 2 (ACE2), thus interfering with the ACE2 function as an entry receptor for SARS-CoV-2.

Altogether, we conclude that the use of high melatonin doses can be problematic in terms of specificity and engagement of molecular targets, two aspects that are rarely addressed but need to be clarified based on experimental evidence (see also the discussion and recommendations in Cecon et al. [9]).

## 9. The Indole Hypothesis

As we detailed previously, particularly in the recommendations on melatonin-related publications [162], the necessity must be emphasized to use the control substances at concentrations similar to melatonin concentrations to evaluate the specificity of an observed effect. This includes the use of other indole-based compounds, including those that may not be oxidized in the same way as melatonin, to address and demonstrate the possible mechanism of action of melatonin for the particular outcome.

Concerning indole-based compounds, it is interesting to evaluate some of the examples that addressed the question of specificity: (a) serotonin was found to be the most effective compound for inhibiting amyloid-β peptide aggregation. Almost all the indole compounds tested in this study prevented amyloid-β peptide fibril formation and increased cell viability between 9 and 25%. Melatonin and serotonin were found to be the most active. Moreover, serotonin increased the expression of SIRT-1 and 2, heat shock protein 70, and heme oxygenase activity—as melatonin is reported to do as well—this being a possible mechanism underlying the observed neuroprotective effect [163]. This paragraph, adapted from the abstract of the paper, is a rare break from the melatonin-does-everything system, suggesting that either high concentrations of natural compounds may have beneficial effects on the disease models or that indole-based compounds have the propensity to interfere with proteins due to their chemical natures. (b) Zhai et al. demonstrated that a series of indole-based compounds (including tryptamine and tryptophane but not the unrelated histidine) were able, at the same concentrations (3 mM), to block the virus infectivity, probably via a virus/receptor interaction [30]; see also [161]. (c) Wölfler et al. showed that *N*-acetylserotonin is a better compound for antioxidant activity than melatonin, especially at 10 µM, at which the concentration of melatonin is almost inactive in this cellular model [164]. (d) When the neuroblastoma SK-N-MC cells were treated either by hydrogen peroxide (H_2_O_2_) or following glutamate-induced cell death, *N*-acetyltryptamine, as well as melatonin, were reported to protect the cells against those injuries, although at concentrations from 10 to 500 µM. The authors suggested that the protection occurred via the induction of NF-κB [165]. The systematic inclusion of melatonin-related compounds and the determination of the EC_50_ values are highly desirable in future studies to better understand the effects associated with melatonin, in particular, at high concentrations. From a therapeutic point of view, the putative interchangeability of indole-based compounds, such as melatonin, *N*-acetyltryptamine, etc., would increase the number of therapeutic choices to minimize the harmful effects, depending on the disease context.

## 10. What to do?

For several years now, a growing number of claims can be said to have distracted melatonin research and trivialized the role that endogenously produced melatonin has in maintaining circadian sleep patterns, metabolism and mental health.

Indeed, the disruption of light rhythms in our cities and the constant use of electronic devices with artificial light have led to questions about the way public health is affected by such changes. In the meantime, significant resources—intellectual, financial and societal—are potentially wasted by moving forward the working repetitive claims/hypotheses that are not, or only insufficiently, supported by experimental evidence with all the required scientific rigor. Clearly, only the clarification of the basics and not the extension of non-consensual theories will lead to a general consensus in the scientific community to eventually move on in a coordinated manner and on a solid scientific basis to address the most relevant questions and challenges.

Another way to reduce the impact of misleading claims is to discourage reviews of the literature that simply and uncritically repeat statements or discuss experiments.

Another point should be stressed: the use of melatonin at very high dosages in patients and infants should be restricted as a precautionary principle. Indeed, as pointed out on several occasions, melatonin toxicity, in vivo, remains unexplored territory at these dosages, even at moderate dosages. There have been no rigorous toxicity studies reported in humans, and the repeated claims that a compound being natural cannot be toxic, are potentially harmful.

## Figures and Tables

**Figure 1 biomolecules-13-00943-f001:**
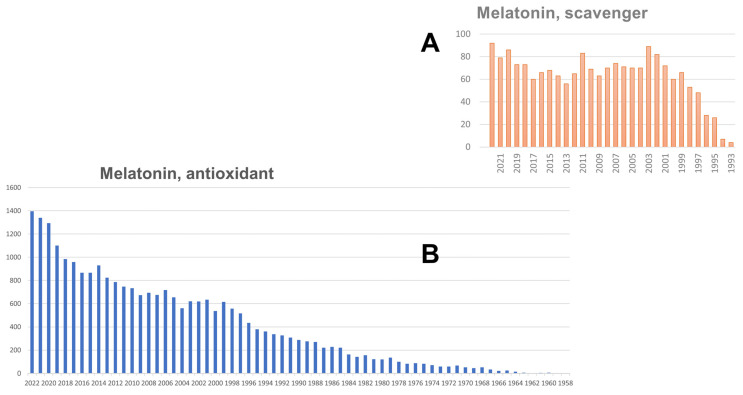
PubMed extraction of queries “melatonin + scavenger” (**A**); and “melatonin + antioxidant” (**B**).

**Table 1 biomolecules-13-00943-t001:** A non-exhaustive list of the many claimed wonders of melatonin.

Pathological Conditions	Title		Exp/Rev	Melatonin Concentration
Aging	Melatonin as an Anti-Aging Therapy	[20]	Rev	
Aging	Protective Role of Melatonin and Its Metabolites in Skin Aging	[21]	Rev	
Alzheimer’s Disease	Mechanisms of Melatonin in Alleviating Alzheimer’s Disease.	[22]	Rev	
Anxiety	Melatonin as a Potential Approach to Anxiety Treatment.	[23]	Rev	
Atherosclerosis	Melatonin-based therapeutics for atherosclerotic lesions and beyond:	[24]	Rev	
Bacteria infection	Melatonin inhibits Gram-negative pathogens	[25]	Exp	1 to 4 mM *
Cancer	Melatonin and cancer suppression: …	[26]	Rev	
Cancer	Melatonin Reverses the Warburg-Type Metabolism	[27]	Exp	3.2 mM
Cancer	Oncostatic activities of melatonin: …	[28]	Rev	
Cardiovascular diseases	Evidence for the Benefits of Melatonin in Cardiovascular Disease	[29]	Rev	
Coronavirus infection	Melatonin and other indoles show antiviral activities	[30]	Exp	3 mM **
Diabetes	Coadministration of Melatonin and Insulin Improves Diabetes-Induced…	[31]	Exp	~40 µM **
Dislipidemia	The Mechanism of Oral Melatonin Ameliorates Intestinal and Adipose Lipid Dysmetabolis	[32]	Exp	0.4 mg/mL ***
Ischemy (cellular)	Melatonin Attenuates Ischemic-like Cell Injury	[33]	Exp	50 µM
Neurodegeneration	…New insights into the role of melatonin in neuronal recovery	[34]	Rev	
Neurodegeneration	Melatonin: Regulation of Biomolecular Condensates in Neurodegenerative Disorders	[35]	Rev	
Parkinson’s Disease	Melatonin and Parkinson Disease: Current Status…	[36]	Rev	
SARS infection	Melatonin use for SARS-CoV-2 infection: …	[37]	Rev	
SARS infection	Melatonin: highlighting its use as a potential treatment for SARS-CoV-2 infection	[38]	Rev	

* Assuming 100% passage in 30 g mice with 4 mL of blood; ** melatonin is not the best inhibitor among the indoles tested; *** of drinking water.

**Table 2 biomolecules-13-00943-t002:** “Main” categories under which the melatonin trials have been conducted.

Abnormalities, Multiple	Colitis	Genetic Diseases, Inborn	Musculoskeletal Abnormalities	REM Sleep Parasomnias
Acute Graft versus Host Disease	Colitis, Ulcerative	Genetic Diseases, X-Linked	Musculoskeletal Diseases	Renal Insufficiency
Acute Kidney Injury	Collagen Diseases	Genital Neoplasms, Female	Musculoskeletal Pain	Renal Insufficiency, Chronic
Acute Lung Injury	Colonic Diseases	Glioma	Myalgia	Reperfusion Injury
Acute Respiratory Distress Syndrome	Colonic Diseases, Functional	Glucose Intolerance	Myocardial Infarction	Respiration Disorders
Adnexal Diseases	Communicable Diseases	Glucose Metabolism Disorders	Myocardial Ischemia	Respiratory Aspiration
Alcoholism	Confusion	Gonadal Disorders	Narcolepsy	Respiratory Distress Syndrome
Alcohol-Related Disorders	Congenital Abnormalities	Growth Disorders	Necrosis	Respiratory Distress Syndrome, Infant
Alzheimer’s Disease	Connective Tissue Diseases	Head and Neck Neoplasms	Necrotizing Enterocolitis	Respiratory Distress Syndrome, Newborn
Anaplastic Astrocytoma	Consciousness Disorders	Head Injuries, Closed	Neoplasm Metastasis	Respiratory Hypersensitivity
Anaplastic Ependymoma	Constriction, Pathologic	Headache	Neoplasms by Histologic Type	Respiratory Tract Diseases
Anaplastic Oligodendroglioma	Coronary Artery Disease	Headache Disorders	Neoplasms, Germ Cell and Embryonal	Respiratory Tract Infections
Aneurysm	Coronary Disease	Headache Disorders, Primary	Neoplasms, Glandular and Epithelial	Respiratory Tract Neoplasms
Angelman Syndrome	Coronaviridae Infections	Heart Diseases	Neoplasms, Nerve Tissue	Retinal Diseases
Anorexia	Coronavirus Infections	Heart Failure	Neoplastic Processes	Rheumatic Diseases
Anxiety Disorders	COVID-19	Hemorrhage	Nervous System Neoplasms	RNA Virus Infections
Apnea	Cranial Nerve Diseases	Heredodegenerative Disorders, Nervous System	Neurobehavioral Manifestations	Schizophrenia
Arrhythmias, Cardiac	Craniocerebral Trauma	Hot Flashes	Neurocognitive Disorders	Schizophrenia Spectrum and Other Psychotic Disorders
Arterial Occlusive Diseases	Craniofacial Abnormalities	Hyperglycemia	Neurodegenerative Diseases	Sclerosis
Arteriosclerosis	Critical Illness	Hyperinsulinism	Neurodevelopmental Disorders	Seasonal Affective Disorder
Arthritis	Cysts	Hyperkinesis	Neuroectodermal Tumors	Seizures
Asphyxia	Death	Hyperlipidemias	Neuroectodermal Tumors, Primitive	Sensation Disorders
Asphyxia Neonatorum	Deglutition Disorders	Hyperlipoproteinemias	Neuroendocrine Tumors	Sepsis
Asthma	Delirium	Hyperphagia	Neuroepithelioma	Shock
Astrocytoma	Dementia	Hyperplasia	Neurologic Manifestations	Shock, Septic
Atrophy	Demyelinating Autoimmune Diseases, CNS	Hypersensitivity	Neuromuscular Diseases	Signs and Symptoms, Digestive
Attention Deficit and Disruptive Behavior Disorders	Demyelinating Diseases	Hypersensitivity, Immediate	Neurotoxicity Syndromes	Signs and Symptoms, Respiratory
Attention Deficit Disorder with Hyperactivity	Depression	Hypertension	Nevi and Melanomas	Skin Diseases
Autism Spectrum Disorder	Depression, Postpartum	Hyperthermia	Nevus	Skin Diseases, Eczematous
Autistic Disorder	Depressive Disorder	Hypothermia	Nevus, Pigmented	Skin Diseases, Genetic
Autoimmune Diseases	Depressive Disorder, Major	Hypoxia	Nidovirales Infections	Skin Manifestations
Autoimmune, Diseases of the Nervous System	Dermatitis	Hypoxia, Brain	Non-24-Hour Sleep-Wake Disorder	Sleep Apnea Syndromes
Autonomic Nervous System Diseases	Dermatitis, Atopic	Hypoxia-Ischemia, Brain	Nutrition Disorders	Sleep Apnea, Obstructive
Basal Ganglia Diseases	Developmental Disabilities	Immune System Diseases	Obesity	Sleep Deprivation
Behavioral Symptoms	Diabetes Complications	Infant, Newborn Diseases	Obstetric Labor Complications	Sleep Disorders, Circadian Rhythm
Bipolar and Related Disorders	Diabetes Mellitus	Infant, Premature, Diseases	Obstetric Labor, Premature	Sleep Disorders, Intrinsic
Bipolar Disorder	Diabetes Mellitus, Type 1	Infarction	Occupational Diseases	Sleep Initiation and Maintenance Disorders
Blindness	Diabetes Mellitus, Type 2	Infections	Oculo-cerebral Syndrome With Hypopigmentation	Sleep-Wake Disorders
Body Temperature Changes	Diabetic Angiopathies	Infertility	Oligodendroglioma	Sleepiness
Body Weight	Diabetic Retinopathy	Infertility, Female	Orthostatic Intolerance	Smith-Magenis Syndrome
Body Weight Changes	Digestive System Diseases	Inflammation	Osteoporosis	Spinal Cord Diseases
Bone Diseases	Digestive System Neoplasms	Inflammatory Bowel Diseases	Ovarian Cysts	Spinal Cord Injuries
Bone Diseases, Metabolic	Disease Attributes	Insulin Resistance	Ovarian Diseases	ST Elevation
Brain Concussion	Disease Progression	Intellectual Disability	Overnutrition	Myocardial Infarction
Brain Diseases	Disorders of Excessive Somnolence	Intestinal Diseases	Overweight	Stomatitis
Brain Diseases, Metabolic	Dyskinesias	Intracranial Hemorrhages	Pain	Stomatognathic Diseases
Brain Infarction	Dyslipidemias	Irritable Bowel Syndrome	Pain, Post-operative	Stomatognathic System
Brain Injuries	Dyspepsia	Ischemia	Paralysis	Abnormalities
Brain Injuries, Traumatic	Dyssomnias	Ischemic Stroke	Parasomnias	Stress Disorders, Traumatic
Brain Ischemia	Eczema	Jet Lag Syndrome	Parkinson’s Disease	Stress, Psychological
Brain Neoplasms	Emergence Delirium	Joint Diseases	Parkinsonian Disorders	Stroke
Breast Diseases	Emergencies	Kidney Diseases	Pathological Conditions, Anatomical	Substance-Related Disorders
Breast Neoplasms	Encephalomyelitis	Kidney Failure, Chronic	Pediatric Obesity	Syndrome
Bronchial Diseases	Endocrine System Diseases	Lens Diseases	Periodontal Diseases	Synovial Cyst
Burning Mouth Syndrome	Endometriosis	Lipid Metabolism Disorders	Periodontitis	Synucleinopathies
Burns	Enterocolitis	Liver Cirrhosis	Personality Disorders	Systemic Inflammatory Response Syndrome
Cachexia	Enterocolitis, Necrotizing	Liver Diseases	Photophobia	Tauopathies
Calcinosis	Ependymoma	Lower Urinary Tract Symptoms	Pneumonia	Thoracic Neoplasms
Calcium Metabolism Disorders	Epilepsy	Lung Diseases	Pneumonia, Viral	Tooth Diseases
Carcinoma	Esophageal Diseases	Lung Diseases, Obstructive	Poisoning	Toxemia
Carcinoma, Non-Small-Cell Lung	Esophageal Motility Disorders	Lung Injury	Polycystic Ovary Syndrome	Trauma, Nervous System
Cardiomyopathies	Esophageal Spasm, Diffuse	Lung Neoplasms	Post-operative Complications	Travel-Related Illness
Carotid Artery Diseases	Essential Hypertension	Mania	Precancerous Conditions	Tuberous Sclerosis
Carotid Stenosis	Eye Diseases	Maxillofacial Abnormalities	Prediabetic State	Tuberous Sclerosis Complex
Cataract	Familial Alzheimer Disease	Melanoma	Pregnancy Complications	Ulcer
Central Nervous System Diseases	Fatigue	Menopause	Premature Birth	Urogenital Neoplasms
Central Nervous System Infections	Fatigue Syndrome, Chronic	Menstruation Disturbances	Primary Dysautonomias	Urologic Diseases
Central Nervous System Neoplasms	Feeding and Eating Disorders	Mental Disorders	Problem Behavior	Urological Manifestations
Cerebral Infarction	Fetal Diseases	Metabolic Diseases	Prostatic Diseases	Uterine Cervical Diseases
Cerebral Palsy	Fetal Growth Retardation	Metabolic Syndrome	Pruritus	Uterine Cervical Neoplasms
Cerebrovascular Disorders	Fever	Migraine Disorders	Psychomotor Agitation	Uterine Diseases
Chemically-Induced Disorders	Fibrosis	Mood Disorders	Psychomotor Disorders	Uterine Neoplasms
Child Development Disorders, Pervasive	Flaviviridae Infections	Mouth Diseases	Psychotic Disorders	Vascular Diseases
Chromosome Disorders	Fractures, Bone	Movement Disorders	Puerperal Disorders	Vascular System Injuries
Chronic Graft Versus Host Disease	Frailty	Mucinoses	Quadriplegia	Virus Diseases
Chronic Pain	Ganglion Cysts	Mucositis	Quality of Life	Vision Disorders
Chronic Periodontitis	Gastroenteritis	Multiple Sclerosis	Radiation Injuries	Wasting Syndrome
Chronobiology Disorders	Gastroesophageal Reflux	Multiple Sclerosis, Relapsing-Remitting	Radiodermatitis	Weight Gain
Cognition Disorders	Gastrointestinal Diseases	Muscular Atrophy	Recurrence	Weight Loss
Cognitive Dysfunction	Gastrointestinal Neoplasms	Muscular Diseases	REM Sleep Behavior Disorder	

**Table 3 biomolecules-13-00943-t003:** Each category corresponds to a single trial for melatonin effect(s).

Acanthosis Nigricans	Chromosome Deletion	Hepatitis	Marijuana Abuse	Prognathism
Acid-Base Imbalance	Chronic Disease	Hepatitis A	Melanosis	Prostatic Hyperplasia
Acidosis	Cluster Headache	Hepatitis C	MELAS Syndrome	Prostatic Neoplasms
Acquired Immunodeficiency Syndrome	Colic	Hepatitis C, Chronic	Memory Disorders	Psychophysiologic Disorders
ACTH-Secreting Pituitary Adenoma	Colonic Neoplasms	Hepatitis, Chronic	Mental Retardation, X-Linked	Rare Diseases
Acute Coronary Syndrome	Colorectal Neoplasms	Hepatitis, Viral, Human	Metabolism, Inborn Errors	Retinal Degeneration
Acute Mountain Sickness	Communication Disorders	Herpes Genitalis	Microvascular Angina	Retrognathia
Adamantinoma	Constipation	Herpes Simplex	Migraine with Aura	Rupture
Adrenal Gland Diseases	Contusions	Herpesviridae Infections	Migraine without Aura	Salivary Gland Diseases
Adrenal Insufficiency	Craniopharyngioma	Hip Fractures	Monosomy	Sarcopenia
Adrenocortical Hyperfunction	Crohn Disease	Hip Injuries	Mouth Neoplasms	Scoliosis
Aggression	Cushing Syndrome	HIV Infections	Mouth, Edentulous	Seizures, Febrile
Alcohol Drinking	Delayed Emergence from Anesthesia	Hodgkin Disease	Multiple Myeloma	Self-Injurious Behavior
Alternating Hemiplegia of Childhood	Dengue	Hodgkin Lymphoma	Multiple System Atrophy	Septo-Optic Dysplasia
Altitude Sickness	Dengue Fever	Huntington Disease	Myocardial Reperfusion Injury	Septo-optic Dysplasia Spectrum
Alveolar Bone Loss	Dentofacial Deformities	Hyperadrenalism	Myofascial Pain Syndromes	Severe Acute Respiratory Syndrome
Ameloblastoma	Depressive Disorder, Treatment-Resistant	Hyperandrogenism	Narcotic-Related Disorders	Sex Chromosome Disorders
Amino Acid Metabolism, Inborn Errors	Diabetes Insipidus	Hyperpigmentation	Neonatal Sepsis	Sexually Transmitted Diseases
Anaplastic Oligoastrocytoma	Diabetes Insipidus, Neurogenic	Hypertension, Portal	Neoplasms, Neuroepithelial	Sexually Transmitted Diseases, Viral
Aneuploidy	Diarrhea	Hypertension, Pregnancy-Induced	Neoplasms, Plasma Cell	Shy-Drager Syndrome
Anodontia	Diffuse Large B-Cell Lymphoma	Hypo-hidrotic Ectodermal Dysplasia	Neoplasms, Squamous Cell	Skin Abnormalities
Anorexia Nervosa	DNA Virus Infections	Hypokinesia	Nervous System Malformations	Skin Diseases, Infectious
Aortic Aneurysm	Drug-Resistant Epilepsy	Hypopituitarism	Neuromuscular Manifestations	Skin Diseases, Viral
Aortic Aneurysm, Abdominal	Drug-Related Side Effects and Adverse Reactions	Hypotension	Night Eating Syndrome	Somatoform Disorders
Aortic Diseases	Dysbiosis	Hypotension, Orthostatic	Nocturia	Speech Disorders
Aphasia	Dyskinesia, Drug-Induced	Hypothalamic Diseases	Nocturnal Enuresis	Spinal Curvatures
Arbovirus Infections	Dysmenorrhea	Hypothalamic Obesity	Obesity, Morbid	Spinal Diseases
Arthritis, Juvenile	Ear Diseases	Idiopathic Hypersomnia	Oligoastrocytoma	Squamous Cell Carcinoma of Head and Neck
Arthritis, Rheumatoid	Ectodermal Dysplasia	Immunoproliferative Disorders	Opioid-Related Disorders	Stillbirth
Atrial Fibrillation	Ectodermal Dysplasia 1, Anhidrotic	Inborn Amino Acid Metabolism Disorder	Optic Nerve Diseases	Stress Disorders, Traumatic, Acute
B-cell Lymphoma	Emaciation	Influenza, Human	Optic Nerve Hypoplasia	Substance Withdrawal Syndrome
Back Pain	Endotoxemia	Intestinal Neoplasms	Oral Cancer	Sunburn
Bacteremia	Enterovirus Infections	Intracranial Aneurysm	Oral Leukoplakia	Tachycardia
Barrett Esophagus	Enuresis	Intracranial Arterial Diseases	Oral Squamous Cell Carcinoma	Tachycardia, Sinus
Binge-Eating Disorder	Epilepsy, Generalized	Intracranial Hemorrhage, Traumatic	Osteoarthritis	Tardive Dyskinesia
Birth Weight	Epileptic Syndromes	Intraocular Melanoma	Osteoarthritis, Knee	Thoracic Injuries
Blood-Borne Infections	Erythema	Jaw Abnormalities	Osteoporosis, Postmenopausal	Tic Disorders
Bone Neoplasms	Eye Diseases, Hereditary	Jaw Diseases	Otorhinolaryngologic Diseases	Tics
Bone Resorption	Eye Neoplasms	Lacerations	Pancreatic Cancer	Tinnitus
Borderline Personality Disorder	Facies	Language Disorders	Pancreatic Neoplasms	Tobacco Use Disorder
Brain Damage, Chronic	Femoral Fractures	Leg Injuries	Pelvic Pain	Tooth Abnormalities
Brain Diseases, Metabolic, Inborn	Fetal Alcohol Spectrum Disorders	Lennox Gastaut Syndrome	Periodontal Atrophy	Tooth Loss
Bronchial Neoplasms	Fetal Death	Leukoencephalopathies	Phenylketonurias	Tooth, Impacted
Bronchitis	Fetal Membranes, Premature Rupture	Leukoplakia	Photosensitivity Disorders	Tourette Syndrome
Bulimia	Fibromyalgia	Leukoplakia, Oral	Picornaviridae Infections	Trauma and Stressor Related Disorders
Bulimia Nervosa	Flavivirus Infections	Lewy Body Disease	Pigmentation Disorders	Trigeminal Autonomic Cephalalgias
Burnout, Psychological	Flushing	Lightning Injuries	Pinealoma	Unconsciousness
Carcinogenesis	Fragile X Syndrome	Liver Failure	Pineocytoma	Urinary Incontinence
Carcinoma, Bronchogenic	Genital Neoplasms, Male	Low Back Pain	Pituitary ACTH Hypersecretion	Urination Disorders
Carcinoma, Squamous Cell	Headache Disorders, Secondary	Lymphatic Diseases	Pituitary Diseases	Uterine Hemorrhage
Cardiac Conduction System Disease	Hearing Disorders	Lymphoma	Post-Concussion Syndrome	Uveal Diseases
Caregiver Burden	Heat Stress Disorders	Lymphoma, B-Cell	Post-operative Cognitive Complications	Uveal Neoplasms
Central Diabetes Insipidus	Hematoma	Lymphoma, Large B-Cell, Diffuse	Postpartum Hemorrhage	Vector Borne Diseases
Cerebral Hemorrhage	Hematoma, Subdural	Lymphoma, Non-Hodgkin	Postural Orthostatic Tachycardia Syndrome	Viral Hemorrhagic Fever
Childhood Acute Lymphoblastic Leukemia	Hematoma, Subdural, Chronic	Lymphoproliferative Disorders	Prader-Willi Syndrome	Vision, Low
Chorea	Hemiplegia	Lymphosarcoma	Prehypertension	X-linked Hypo-hidrotic Ectodermal Dysplasia
Chorioamnionitis	Hemorrhagic Fevers, Viral	Macular Degeneration	Premenstrual Dysphoric Disorder	Xerostomia
Chromosome 17p Deletion	Hepatic Encephalopathy	Malocclusion, Angle Class III	Premenstrual Syndrome	
Chromosome Aberrations	Hepatic Insufficiency	Mandibular Diseases	Primary Orthostatic Hypotension	

**Table 4 biomolecules-13-00943-t004:** Details of the currently recruiting trials on melatonin.

Conditions	Number Patients	Treatment	Duration (Days) (Years)	Observational	
Smith-Magenis syndrome	8		1	Mel measure	NCT02180451
Autism	105		3	Mel measure	NCT02878499
Cushing disease	15		90	Mel measure	NCT03343470
Breast Neoplasms	27		270	Mel measure	NCT04364347
Alzheimer’s Disease	60		*	Mel measure	NCT04522960
Sleep deprivation	20		32	Mel measure	NCT04868539
Frail Elderly	300		3	Mel measure	NCT05107947
Hypo-hidrotic Ectodermal Dysplasia	80		10	Mel measure	NCT05378932
Bipolar disorder	80		2	Mel measure	NCT05413486
Alzheimer’s Disease	164		42	Mel measure	NCT05543681
Sleep Disturbance	60		3	Mel measure	NCT05647148
Hypoxic-ischemic Encephalopathy	70	0.5–5 mg	14		NCT02621944
Delirium	190	4 mg	28		NCT03438526
Multiple Progressive Primary Sclerosis	50	100 mg daily	2		NCT03540485
Alzheimer’s Disease	230	5 mg daily	231		NCT03954899
Osteopenia	40	5 mg daily	1		NCT04233112
Attention Deficit Hyperactivity Disorder	80	3 mg daily	0.5		NCT04318067
Postoperative delirium	790	4 mg daily	10		NCT04335968
Huntington’s Disease	20	5 mg daily	63		NCT04421339
COVID-19 infection	30	3 × 10 mg daily	28		NCT04474483
Diabetic Eye Problems	36	3 mg daily	14		NCT04547439
Post-operative pain	60	10 mg 3 days	21		NCT04791943
Osteoarthritis	252	5 mg daily	60		NCT04795336
Traumatic Brain Injury	110	3–5 mg daily	30		NCT04932096
Necrotizing Enterocolitis	100	6 mg daily	150		NCT05033639
Emergence Agitation	117	5 mg pre-op ***	1		NCT05223010
Ischemic Stroke	80	3 mg daily	14		NCT05247125
Hypertension	23	1 mg daily	1		NCT05257291
PD	50	5 mg daily	28		NCT05307770
Epilepsy	120	5 mg daily	28		NCT05439876
Post-menopause insomnia	14	2 mg daily	15		NCT05440734
Chronic Fatigue Syndrome	106	1 mg daily	28		NCT05454683
Uveal Melanoma	100	20 mg daily	5		NCT05502900
Severe Preterm Fetal Growth Restriction	336	3 × 10 mg daily **	126		NCT05651347

* From 2 days to 2 years. ** Treatment to the mother, not clearly stating for how many days.*** Pre-op means pre-operation.

**Table 5 biomolecules-13-00943-t005:** Bacteria that produce melatonin, as reported by Que et al. [92].

Bacteria Type	Reference	Comments (See Text)
* Cyanobacteria, *	[95]	Comment A
* Rhodospirillum rubrum *	[95,96]	Comments A & B
*Bacillus cereus* CS-17	[97]	Comment C
*Ensifer* sp. VA11,	[97]	Comment C
*Pseudomonas* sp.	[97]	Comment C
*Variovorax* sp.	[97]	Comment C
*Agrobacterium tumefaciens*	[97]	Comment C
*Bacillus amyloliquefaciens*	[97]	Comment C
*Bacillus thuringiensis*	[97]	Comment C
*Sphingomonas* sp.	[97]	Comment C
*Bifidobacterium breve*,	[98]	Comment D
*Lactobacillus brevis*,	[98]	Comment D
*Lactobacillus casei,*	[98]	Comment D
*Bifidobacterium longum*,	[98]	Comment D
*Enterococcus faecalis* TH10,	[98]	Comment D
*Lactobacillus acidophilus*,	[98]	Comment D
*Lactobacillus bulgaricus*,	[98]	Comment D
*Lactobacillus fermentum*,	[98]	Comment D
*Lactobacillus helveticus*	[98]	Comment D
* Lactobacillus plantarum * ,	[98]	Comment D
* Streptococcus thermophilus, *	[98]	Comment D
* Erythrobacter longus * ,	[99]	Comment E

Color code: **green**: melatonin synthesis proven; **orange**: melatonin synthesis claimed (by patents); **red**: absence of melatonin synthesis reported.

## Data Availability

Not applicable.

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
