# Peer review of "Melatonin: Facts, Extrapolations and Clinical Trials"

_biomolecules, 2023, doi:10.3390/biom13060943_

Round 1
Reviewer 1 Report
This is a well written MS, which critically reviews the literature related to melatonin, including several usual claims which frequently appear as the final points in many papers investigating melatonin effects.
These authors critically consider the wishful thinking and hypothesis explaining positive effects of administered melatonin in nearly all pathologies. I agree with their conclusion that most claims related to melatonin remain hypotheses and a critical assessment of the existing literature on a role of melatonin as an antioxidant and scavenger in clinical context is needed. Many reviews on melatonin are available now, but the focus on clinical studies is an important advantage of this one. However, several issues must be answered before the MS is accepted for publication.
Comment 1.
The authors are well known in the field of melatonin measurement and melatonin receptor and probably this is why they focus predominantly on receptor-mediated, mechanism of action of melatonin administered in high pharmacological doses. The action does not need to be the same, as mentioned by another leading person (Joe Arendt) in melatonin research - “In principle the established role of melatonin in rhythmic function is not necessarily incompatible with the use of high doses for ‘protective’ effects.” (Arendt J.: Melatonin: countering chaotic time cues. Front Endocrinol., Front in Endocrinol. 10:391, 2019). https://doi.org/10.3389/fendo.2019.00391). Please, comment.
Comment 2.
The main concern of the authors lies in the underestimation of the possible negative effects of melatonin during pregnancy and early ontogeny, although there is a general consent on low or no toxicity of this compound. They mention a separate recent study reporting undetermined deaths of infants and toddlers with high exogenous melatonin as a result of deliberate or incidental ingestion of melatonin [49; https://doi.org/10.1093/jat/bkac033]. In given context it seems that melatonin was a directly involved, but this is not the case. Please explain more.
In contrast to conclusions of this MS, another recent review (Vine et al., Melatonin use during pregnancy and lactation: A scoping review of human studies of clinical studies Braz J Psychiatry. 2022 May-Jun; 44(3): 342–348, doi: 10.1590/1516-4446-2021-2156) concludes that “clinical studies to date suggests that melatonin use during pregnancy and breastfeeding is probably safe in humans and emphasizes the need for clinical studies…, including exogenous melatonin, during pregnancy and lactation. Please, discuss these different opinions.
Comment 3. The same is true for a note at pp 131-132 „the alert put forward in a possible link between melatonin and infant sudden death syndrome [49]“ This is a very important claim, which requires additional information and discussion here on the basis of case report and not just a reference.
Comment 4 Are the adverse effects, mentioned in part 124- 132 and based on “somewhat patchy data collection via poison centers receiving information on adverse events (www.poison.org) documented in registered trials? Please, explain and discuss.
Comment 5.
214-217 It is known that melatonin is eliminated efficiently by the liver. Is the elimination dose- dependent? Was this proved also with high doses of melatonin mentioned in studies discussed in next paragraphs?
Comment 9
pp. 220-221 Authors mention “Melatonin was generally used at concentrations higher than 1 μM – and even 1 mM. Thus, those concentrations are not expected to be reached in living animals or in human. However, a drug does not need to follow physiological concentrations, especially if the receptor mediated effect is not expected.
Comment 10
In the context with the above-mentioned comment, I appreciate discussion and the suggested hypotheses, especially the indole hypothesis. The fact, that other indoles, such as N-acetyltryptamine, can protect the cells against injuries in the same way as melatonin, proves the protective role of melatonin, which does not require receptor-mediated mechanism. This may explain the need of high doses of compounds for protective effects. In this context serotonin or N-acetyltryptamine can have much more harmful effects than melatonin itself. Please, discuss.
Minor:
Table 2 should be better explained to be understandable for readers.
Author Response
cf file

Reviewer 2 Report
This is a timely review on melatonin, but justifications for its claims require much more published support. The authors have raised some questions and concerns and these may be justified. However, this review suffers from a serious lack of depth and merely cherry-picks data to fit their ideas, as well as unbalanced data presentation. There are papers in every field that present data contrary to the norm. Generally, the authors seem not to have a basic knowledge of the field of melatonin research or in other related areas as well. The following issues need reconciliation.
1. That melatonin is only synthesized in pineal gland is no longer a tenable concept. Virtually all cells, tissues and organs tested may have the ability to synthesize melatonin. Invertebrates, and plants, all of which lack a pineal gland, have measurable amounts of melatonin. Also, certainly the authors cannot deny the synthesis of melatonin in the gut, brain or skin. These observations have been confirmed in countless studies and are widely supported experimentally. While the very high levels of melatonin in the cerebrospinal fluid are probably derived from the pineal gland after its release into the pineal recess (Melatonin enters the cerebrospinal fluid through the pineal recess. Tricoire H, Locatelli A, Chemineau P, Malpaux B. Endocrinology. 2002 Jan;143(1):84-90.), it would be difficult to explain the high concentrations of melatonin in other bodily fluids if the authors content the major melatonin is only of pineal origin. See, for example, data from the bile (High physiological levels of melatonin in the bile of mammals 1999 Tan D-X, Chen L-D, Poeggeler B,. Manchester L.C, Reiter R.J.) the milk (A rapid method to measure melatonin in biological fluids (milk and serum) and other fluids). These values were determined using liquid chromatography-tandem mass spectrometry. Yao S, Liu X, Fu Y, Guan S, Liu Y, Yan L, He P, Liu G. Food Chem. 2023 Mar 15;404(Pt A):134606.) as were the melatonin levels in plants. If the authors maintain majority of melatonin is only of pineal origin, they must provide an explanation for the thousands of papers that report otherwise. It is inadequate to merely ignore the data or deny their validity.
2. The initial publication of melatonin as a free radical scavenger is not Poeggeler et al. [67] as mentioned by the authors. If the authors carefully read the title they should know that it is a hypothesis. The original paper on this topic is "Melatonin: a potent, endogenous hydroxyl radical scavenger. Tan DX, Chen LD, Poeggeler B, Manchester L C, Reiter RJ ". Endocr. J. 1993. 1: 57–60.” which is a research article. Relative to the question about melatonin’s scavenging ability. The Tan et al report used electron spin resonance spectroscopy to document the scavenging activity of melatonin; this is agreed upon to be the most definitive test to document a radical scavenger.
3. Antioxidant enzymes are only one component of the antioxidant network. There are literally hundreds of radical scavengers including vitamins C and E, glutathione, polyphenols, anthocyanins, etc., that participate in reducing oxidative stress.
4. Melatonin is qualified as free radical scavenger and countless research papers have confirmed this point and currently many hundreds of papers make this claim in the interpretation of their data. The paragraph “Scavengers do not exist in complex media, living systems, etc. True antioxidant defense is provided by enzymes that remove superoxide and hydrogen peroxide with rate constants that are hundreds of thousands to more than a million times greater than that of melatonin. For hydroxyl radical, no compound is an effective scavenger and antioxidant defense involves prevention of its formation rather than its removal see discussions in [70], in [71] and in [72]” indicate a misunderstanding of the concept of antioxidant activity and system. As mentioned above, antioxidant enzymes are just one component of the antioxidant system. Molecular scavengers are surely part of the “true” antioxidant system and are involved in antioxidant defense. Yes, melatonin may be less efficient as a scavenger of superoxide and hydrogen peroxide than SOD or catalase, but this is not justification to ignore it as a radical scavenger since there are many other types of ROS and RNS in addition to the superoxide and hydrogen peroxide. Another point is the differential distribution and concentration of melatonin within cellular compartments and the fact that the effective concentration of melatonin is increased since the metabolites of melatonin are also scavengers. As to hydroxyl radical, it is better to prevent its formation, but this is difficult to achieve it. The majority of the oxidative damage (up to 60%) is caused by this species. Once it is generated, there is no enzyme that scavenges it since its reaction rate surpasses any enzyme activity rate; therefore, it requires an antioxidant with diffusion-limited rate. Melatonin is reported to be one of these antioxidants and its reaction with hydroxyl radical reaches k(r) = 2.7 x 1010 M-1 s-1, which is a very high diffusion-limited rate. See publications related to the reaction of melatonin and related indoles with hydroxyl radicals: ESR and spin trapping investigations; as one example see. Matuszak Z, Reszka K, Chignell CF. Free Radic Biol Med. 1997;23(3):367-72. doi: 10.1016/s0891-5849(96)00614-4). In addition, countless studies have documented the greater scavenging activity of melatonin than the classic antioxidant vitamin C , E and glutathione. These should be mentioned to balance the discussion.
5. As to the bacterial melatonin, as listed in the table 5, five published papers have identified bacteria that produce melatonin and 12 patents have claimed bacteria synthesize melatonin. The data indicate that the majority of the tested bacteria have the capacity to produce melatonin; based on the data there is no reason to believe that melatonin cannot be synthesized by any of these species. Interestingly, the authors claimed that “out of the reported 30,000 bacteria on Earth, to the best of our knowledge there are only reports on 5 of them producing melatonin”. Are the authors claiming that the investigators were merely fortunate enough to select bacteria for melatonin measurement and just happen to select those that do? If that argument is used then also, there are 65,000 vertebrate species on earth; less than a few hundred have been used to measure their melatonin. Yet, there seems to be wide agreement that vertebrates (likely all) synthesize melatonin. The authors seem to imply that until all 30,000 bacteria are analyzed, claims should not be made about the percentage that synthesize melatonin? This is an unreasonable precaution. Recently, new evidence definitely prove the another bacterial species to produce melatonin (Molecular Cloning and Characterization of a Serotonin N-Acetyltransferase Gene, xoSNAT3, from Xanthomonas oryzae pv. oryzae. Chen X, Zhao Y, Laborda P, Yang Y, Liu F. Int J Environ Res Public Health. 2023 Jan 19;20(3):1865. doi: 10.3390/ijerph20031865.) or gut microbiota (Gut microbiota-derived melatonin from Puerariae Lobatae Radix-resistant starch supplementation attenuates ischemic stroke injury via a positive microbial co-occurrence pattern. Lian Z, Xu Y, Wang C, Chen Y, Yuan L, Liu Z, Liu Y, He P, Cai Z, Zhao J. Pharmacol Res. 2023 Apr;190:106714. doi: 10.1016/j.phrs.2023.106714. Epub 2023 Feb 28). Please update the information.
6. There are hundreds or even thousands of papers that have reported scavenging and antioxidant activities of melatonin in plants. This strongly supports melatonin as an antioxidant in organisms in general. In addition, extremely high melatonin has been detected in plants and plant products. The authors seem to lack knowledge of these developments. To improve the quality of the review, the authors must explain why all these hundreds of reports are incorrect.
7. In general, the authors seem to be concerned about too many research and reports on melatonin as an antioxidant and melatonin being used to treat different diseases. Actually, this is an excellent and natural trend; this is now discoveries of effective drugs (even those that are very toxic) are made. Not to test very low toxicity melatonin for the same reasons would be unscientific. The more research and more clinical trials will result in further proof or disproof of these hypotheses or concepts. Only the increased experimentation can separate the “wheat from the chaff” (“or the wishful thinking from the facts”) as mentioned by the authors.
8. In conclusion, it is always possible to find published literature that is contrary to data presented in thousands of publications. The current report is an example of this. The authors may want to rethink some of their claims or back them up with further evidence. It is fine to be an iconoclast, but only if one has the necessary support/proof to justify it.
English is fine
Author Response
cf file

Round 2
Reviewer 1 Report
Authors answered my questions.
I have a minor additional comment. Please, be more careful with discussion of rerotonin administration and do not include serotonin in the last part of your MS (677-680) you have added. Cardiovascular toxicity of serotonin is well known as well as the pathology (carcinoid) resulting from an overproduction of serotonin in the gut.
The fascinated aspect of melatonin is the absence of such clear and well proved negative consequences of melatonin administration even in such hight doses as mentioned in this review.
Author Response
Thank you for pointing out this to us.
Of course, the word "serotonin" was withdrawn from line 678.